# Butterfly Communities Vary under Different Urbanization Types in City Parks

**DOI:** 10.3390/ani13111775

**Published:** 2023-05-26

**Authors:** Ying Lin, Shanjun Huang, Wenqiang Fang, Yujie Zhao, Ziluo Huang, Ruoxian Zheng, Jingkai Huang, Jiaying Dong, Weicong Fu

**Affiliations:** 1College of Landscape Architecture, Fujian Agriculture and Forestry University, 15 Shangxiadian Road, Fuzhou 350002, China; 2School of Architecture, Clemson University, 105 Sikes Hall, Clemson, SC 29634, USA; 3Engineering Research Center for Forest Park of National Forestry and Grassland Administration, Fuzhou 350002, China; 4Collaborative for Advanced Landscape Planning, The University of British Columbia, Vancouver, BC V6T 1Z4, Canada

**Keywords:** biodiversity, urbanization, butterfly distribution, urban parks

## Abstract

**Simple Summary:**

Urban biodiversity conservation is currently a prominent issue in society. Butterflies serve as excellent environmental indicator species, and enhancing butterfly diversity can significantly enhance the quality of urban habitats. To contribute to this field, we conducted an analysis of butterfly diversity in various urban gradients. Our research incorporated Shannon diversity analysis, β-diversity analysis, familial diversity analysis, and indicator species analysis. We identified the characteristics and patterns of butterfly diversity distribution and aimed to provide useful insights for urban builders.

**Abstract:**

Butterflies are key indicators of urban biodiversity and one of the most vulnerable organism groups to environmental changes. Studying how butterflies are distributed and what factors might influence them in urban green spaces is crucial. In this study, from July 2022 to September 2022, we examined and analyzed the butterfly diversity in nine parks in Fuzhou, China, along three different levels of urbanization (urban, peri-urban, and suburban). We investigated how butterfly communities respond to increasing urbanization. The findings revealed that: (1) A total of 427 butterfly individuals from 4 families and 13 species were observed; (2) Shannon diversity, richness, and abundance of the overall butterfly community were lower in the more urbanized parks. Urbanization had significant effects on Shannon diversity (*p* = 0.003) and abundance (*p* = 0.007) but no significant effects on the whole butterfly community richness (*p* = 0.241); (3) non-metric multidimensional scaling revealed that there were differences in the overall number of butterfly species in urban parks among different geographic regions.

## 1. Introduction

Although urbanization has raised the public standard of living materially and spiritually [1], it has also caused side effects such as ecological degradation, a loss of biodiversity, and a rise in landscape fragmentation [2]. The physical environmental conditions of the land have altered as a result of urbanization, leading to a significant impact on the habitat and environment of biological species [3]. The massive transformation of urban ecosystems, the invasion and fragmentation of natural habitats, and the global increasing extreme weather events globally pose a major threat to urban biodiversity [4]. Urban biodiversity conservation and utilization is one of the greatest global challenges. While the conservation of biodiversity in urban, suburban, or peri-urban is addressed in fewer than 6% of papers in “Conservation Biology”, there is significant social and educational value in promoting the conservation and restoration of native habitats in densely populated regions [5]. By protecting biodiversity, urban ecological infrastructure can be enhanced, the landscape can be improved, and a rich living environment can be maintained [6]. Previous studies have shown a strong positive correlation between urban biodiversity and habitat heterogeneity [7]. Therefore, it is reasonable to argue that increasing urban biodiversity can contribute to creating a healthy urban ecological environment.

Butterflies have been proven to be a reliable environmental indicator species [8,9] due to their sensitivity to natural environmental variations in performance [10,11]. Scientists across the world have studied how local and landscape-level factors affect butterfly communities in an urban context. Koh and Sodhi (2004) concluded that reducing the distance between urban parks and forests and raising the variety of potential host plants could protect the biodiversity of urban parks more effectively [12]. Kadlec et al., (2008) investigated the changes in the composition of butterflies and burnet fauna by repeating a survey that had been conducted three decades prior. They observed that the losses of butterfly species were relatively lower compared to similar surveys conducted elsewhere in Central Europe, which was attributed in part to the high heterogeneity of the urban landscape surrounding the reserves [13]. Öckinger E et al., (2009) highlighted the significance of “townscape” composition in determining species richness in urban habitats; both butterfly species richness and abundance increased with decreasing connectivity [14]. Bergerot et al., (2011) examined the structure and composition of butterfly assemblages across an urbanization gradient by analyzing landscape element variables. The findings of their study showed that in strongly urbanized sites, the mean species richness and mean feeding specialization of adult butterflies were significantly lower compared to less modified sites [15]. Ramírez-Restrepo et al., (2017) compiled publications that specifically investigated urban butterflies (Lepidoptera) and found that the majority of the studies reported a negative correlation between urbanization intensity and butterfly diversity, including both richness and abundance [16]. Aguilera G et al., (2019) concluded that intensive management strategies for urban green spaces negatively affect the butterfly assemblage over time, based on the fact that disturbance and management intensity typically increase with urbanization [17]. Currently, studies on butterfly diversity in China are primarily focused on nature reserves [18], with less research on the effects of urbanization on butterfly communities. Yang, D.R. et al., (1998) studied the structure and diversity of butterfly communities in fragmented tropical rainforests in Xishuangbanna and found that the butterfly communities in primary tropical rainforests exhibited greater diversity, higher species richness, and more evenness compared to fragmented ones [19]. Li Cushan et al., (2007) examined the butterfly ecosystem of Changbaishan Nature Reserve [20]. Zhou Guangyi et al., (2016) investigated butterfly diversity and zonation in Nanling National Nature Reserve [21]. Ferenc M et al., (2014) studied the species richness and β-diversity patterns of urban bird communities and regional species assemblages in Europe [22], which has implications for butterfly research in the Chinese region. Given that the diversity of butterflies and their response of butterfly assemblages to urbanization in rapidly growing cities is still poorly understood, particularly in the southeastern subtropical region of China, which has not been studied, local governments need to understand how urbanization affects the distribution of butterfly communities for effective green space management and decision making.

This investigation was carried out in Fuzhou City, Fujian Province, which is in the southeast coastline region of China. Fuzhou is an important city in the West China Sea region, and its development and planning are helpful to other coastal cities; it is valuable to study urbanization effects in Fuzhou because these results can inform the development of functional green infrastructure in other cities in the area that are also rapidly urbanizing. In this study, the following questions will be addressed: (1) Whether increasing urbanization simplifies butterfly assemblages; (2) whether urban parks with lower levels of urbanization provide more ecological niches for butterfly communities to utilize; (3) whether the intensity of urbanization has an effect on the composition and abundance of butterfly communities; and (4) which butterfly species can be used as indicator species for monitoring the level of urbanization. This paper examined the distribution of butterfly species in urban parks in Fuzhou along various urbanization gradients. It can greatly assist green space administrative staff in planning and designing urban green spaces in a logical manner to enhance the urban ecological environment.

## 2. Materials and Methods

### 2.1. Study Area

Fuzhou, which has a subtropical monsoon climate with an average annual rainfall ranging from 900 to 2100 mm, is situated near the southernmost edge of the Eurasian continent, in the downstream coastal region of the Min River in eastern Fujian Province [23]. Fuzhou is also a historic city that has undergone increasing levels of urbanization, with a rate of 73% urbanization among its resident population as of 2021. Moreover, the city’s population continues to grow steadily, with a recorded population of around 8.3 million in Fuzhou City based on data from the 7th National Census of Fuzhou [24]. The following criteria were used to determine which parks to be included in this research: (1) various geographic locations within the city; (2) a rich composition and structure of the vegetation with a diversity of habitat types; (3) out of a total of 35 parks large than 3 ha, and 9 parks were chosen for the study using a random sampling method: Chating Park (3.57 ha), Liming Lake Park (5.67 ha), Wushan Park (14.68 ha), Yushan Park (7.94 ha), Zuohai Park (30.73 ha), Jinji Mountain Park (42.80 ha), Hot Spring Park (17.30 ha), Jinshan Park (31.64 ha), and Helin Ecological Park (16.67 ha) (Figure 1).

Chating Park and Liming Lake Park both have small bodies of water in their interior, and the combination of tree, shrub, and grass vegetation is more common in the parks. Wushan Park and Yushan Park are centrally-located mountain parks that exhibit a more diverse vegetation composition. Zuohai Park and Hot Spring Park are both riverside parks and are rich in trees, shrubs, and ground cover. Jinji Mountain Park is a larger mountain park with a wider variety and mix of vegetation. Jinshan Park and Helin Ecological Park are both waterfront parks, and both are large ecological parks in Fuzhou with better habitats. These parks selected are dispersed across the city’s various urban gradients, and they all feature easy access to public transit, abundant vegetation, and a variety of butterfly species.

### 2.2. Categorization of Urbanization Patterns

The First Ring Road was built when Fuzhou extended southward in tandem with the city’s continental land expansion throughout the Ming and Qing dynasties. It emerged from the old city in the Ming and Qing dynasties and grew into a commercial port along the river in the south, which facilitated the development of the First Ring Road. Since the Fuzhou port opened in modern China, the commercial traffic on the First Ring Road has grown in richness [25]. As a result, the First Ring Road had the highest population density, road density, and building density. The Second Ring Road, the first expressway encircling Fuzhou, served as the hub of the city’s transportation system. It encompassed Fuzhou’s core business region and nearly reached its maximum land utilization capacity [26]. The inauguration of the Third Ring Road on New Year’s Day in 2012 marked Fuzhou’s official entry into the Third Ring Road era. Since 2013, the Third Ring Road’s channels have been rebuilt, and the surrounding infrastructure has undergone constant improvement [26]. Additionally, the east, west, south, and north sides of the Third Ring Road are close to Gushan, Wuhu Mountain, Qishan, and Lianhua Peak [23]. We optimized the method used in earlier research which classified urbanization in accordance with the First to Third Ring Road plan of Fuzhou [27]. The parks within the Third Ring Road of Fuzhou were divided into (1) urban parks, (2) peri-urban parks, and (3) suburban parks. As a result, Fuzhou’s central urban region was split into three urbanization gradients, ranging from UZ1 to UZ3. UZ1 represents the urban core and area with the highest density, while UZ3 represents the urban edge and territory with the least density. Of the nine parks selected for urbanization classification, four urban parks within the First Ring are Chating Park, Liming Lake Park, Wushan Park, and Yushan Park. These parks were perceived as urban parks because they are in the center of the city and have the greatest degrees of urbanization. Zuohai Park, Jinji Mountain Park, and Hot Spring Park are three parks that are located between the First and Second Ring Roads. These parks were considered peri-urban parks since they are less urbanized. The suburban parks, Jinshan Park and Helin Ecological Park, which are situated between the second and third rings, were determined by their separation from the city center and the surrounding low degree of urbanization.

### 2.3. Sampling Criteria

The sampling proceeded along 100-m-long sampling lines put haphazardly around the city park’s inner roads as the basic unit of the butterfly survey [28]. To eliminate spatial correlation, the smallest distance between sample lines was greater than 100 m, and the sample lines should be distributed as evenly as possible within the park. We employed a systematic sampling technique to pick 10 sample bands at random from each urbanization type in order to ensure the validity of the ensuing statistical study. The sample zones for each urbanization type were urban (Chating Park, *n* = 1; Liming Lake Park, *n* = 1; Wushan Park, *n* = 5; Yushan Park, *n* = 3), peri-urban (Zuohai Park, *n* = 3; Jinji Mountain Park, *n* = 5; Hot Spring Park, *n* = 2), and suburban (Jinshan Park, *n* = 6; Helin Ecological Park, *n* = 4).

### 2.4. Butterfly Survey

Fuzhou summer months last from July to September [29], hence the butterfly survey in this paper was carried out from July 2022 to September 2022, and data were collected three times in the same season to include main phenology aspects, once a month, for a total of three months, to examine the characteristics of butterfly distribution in summer. Samples were collected from plots during the period between 9:00 and 17:00, with temperatures over 20 °C, less than 50% cloud cover, and windspeed less than 20 km/h [28]. Moreover, the survey was based on a modified Pollard walk [30,31]. Three researchers who have more than a year of training in butterfly species identification participated in the survey. During the observations, we recorded all butterfly species within 2.5 m to the left and right of the sample strip as well as within 5 m above and in front of the sample strip while moving steadily over the fixed sample strip [28].

Insect nets were used to capture and identify some of the species of butterflies that could not be observed with the naked eye before they were released. The butterfly species observed at the location were recorded (Figure 2). If a few of the species could not be identified on-site, they were put into triangular paper and brought back for identification in the laboratory [32]. These specimens were then examined for identification using literature [33,34,35].

We use classification into families, following, e.g., [36,37], considering that members of the main families share some life history features. Lycaenidae ranges in size from small to medium (wingspan 1~3 cm) and is nimble and slender [34]. Butterflies from the Pieridae visit cruciferous and leguminous plants [38]. A special class of pigments found in the Papilionidae plays a role in the variety of Papilionidae wing colors as well as numerous biological processes such as predator avoidance and mating selection [39]. Beautiful butterflies in the Nymphalidae like flying and moving quickly in the sunlight [40], and their diverse eating patterns allow them to survive in a range of habitats [41,42,43].

### 2.5. Data Analysis

To investigate the functional redundancy of butterfly ecosystems in various urbanization types in city parks, the familial diversity and variety of indicator species of butterfly communities in different urbanization types in Fuzhou city parks were researched. The statistical analyses were conducted using R 4.2.2 [44]. Prior to conducting further analyses, the normal distribution of species richness and abundance for each urbanization type was tested using the Shapiro–Wilk test [45] with the “mvShapiroTest” package [46]. Abundance refers to the count of individual butterflies observed within each urbanization type, while species richness pertains to the number of butterfly species identified within each urbanization type. The findings revealed that the *p*-values for species richness (W = 0.8958, *p*-value = 0.2288) and species abundance (W = 0.8937, *p*-value = 0.2179) were both higher than 0.05, suggesting that these two variables followed a normal distribution and did not need further transformation.

#### 2.5.1. Shannon Diversity Analysis

Shannon Diversity Index is calculated based on species abundance, which refers to the relative abundance of different species in a sample. The calculation of the Shannon Index takes into account the number of species and the distribution of the abundance of each species, thereby reflecting the diversity of species and their relative abundance in the ecosystem [47]. We applied the “vegan” package [48] to calculate the Shannon Diversity Index. Additionally, we examined the Chao1 index, which estimates species diversity based on species abundance (monotypic, bitopic, etc.), and higher values indicate a larger species variety in the community [49]. The Chao1 index (W = 0.9284, *p*-value = 0.4666) and the Shannon diversity index (W = 0.9083, *p*-value = 0.3042) both followed the normal distribution. The “iNEXT” package [49] was used to plot the species accumulation curves, which can be used to gauge and forecast how community species richness would change as sample sizes rise. This method has been frequently employed in biodiversity surveys to evaluate sample size adequacy and estimate community richness.

Further analysis of the impact of various urbanization types on butterfly diversity was conducted using a general linear model (GLM) with a Poisson error structure [50,51]. Urbanization type was the dependent variable, whereas the independent variable was Shannon diversity. To simulate the effect of urbanization type on the response of butterfly communities, we conducted a quasi-Poisson distribution regression within a general linear model (GLM) using the “lmer” function in the “lme4” package [52]. Multicollinearity was found in this model using the variance inflation factor (VIF). We evaluated the homoscedasticity, overdispersion, and model fit. The significance of the effect of different urbanization types on butterfly species was determined by analysis of variance (ANOVA), Tukey HSD test, and conducted a general linearity hypothesis test using the “multcomp” package [53].

#### 2.5.2. β-Diversity Analysis

Changes in species composition were measured using species dissimilarity (β-diversity) [54]. The Bray–Curtis heterogeneity coefficient is undoubtedly one of the 15 most commonly used coefficients by ecologists when analyzing the multivariate structure of species abundance data. It is considered more suitable than the Euclidean distance for assessing community data [55]. Therefore, with the Bray–Curtis function, we determined and ranked the distances between the overall butterfly community, Lycaenidae, Pieridae, Papilionidae, and Nymphalidae in urban parks at various levels of urbanization. The regression is fitted by least squares, which is based on the sum of the squares of the differences between the ranking distance and the regression prediction distance [56]. The log (X + 1) transformation of the richness data led to the determination of the Bray–Curtis dissimilarity matrix. The equation reads as follows:$$BC_{ij}=1-\frac{2C_{ij}}{S_{i}+S_{j}}$$
where $C_{ij}$ represents the sum of the lesser values for butterfly species that are present in both sites i and j, while $S_{i}$ and $S_{j}$ are the total number of butterflies observed in sites i and j Species variability refers to the difference in the occurrence or absence of a species across urban parks of different urbanization types. Non-metric multidimensional scales (NMDS) [56] were used to analyze the makeup of butterfly communities in urban parks to determine the impact of urbanization type on species variability. The “metaMDS” function was utilized, and the stress value was employed as a gauge of the model’s applicability [48]. In general, a stress value under 0.2 was more reasonable, and the stress level in this study was calculated to be less than 0.2. Under the “vegan” package of the “ANOSIM” function [57], statistical differences were quantified using analysis of similarity (ANOSIM). All butterfly species were ranked by 999 species and then repeated in each butterfly family based on the outcomes of the prior tests. The probability distribution and ANOSIM results were then obtained with the significance levels of pairwise differences.

#### 2.5.3. Familial Diversity and Indicator Species

Familial diversity [58] was used to examine the overlap or redundancy of butterfly roles, as well as the relationship between species richness and Shannon diversity of butterfly communities along three urbanization gradients. These relationships were created using the scatter plot in the “ggplot2” package [59].

Differentiating butterfly species under various gradients of urbanization was performed by examining species indicator values. Indicator species [60] could also explain some variables whose ecological characteristics reflected the environmental conditions. The “indicspecies” package [61] combined the mean abundance and probability of occurrence of species within a secondary group. A species with a high indicator value was found in the majority of the sample groups in that sample group (specificity), and the majority of the quadrat groups in this category contained the species (evenness).

## 3. Results

### 3.1. Overview

During the study, a total of 427 individuals from 13 species and 4 families were counted (Table 1). Three butterfly species were particularly prevalent: *Lampides boeticus* (227 individuals), *Pieris rapae* (59 individuals), and *Catopsilia pomona* (38 individuals). Along the urbanization gradient, the peri-urban area had the highest butterfly abundance, with 210 individuals, followed by suburban (137 individuals) and urban areas (80 individuals). The number of butterfly species was highest in the peri-urban park (13 species), second by suburban (9 species), and then urban areas (8 species). The greatest proportion of butterflies belonged to the Lycaenidae in all three urban gradients. No butterflies listed on the national conservation list were discovered during the study [35] (Table 2).

### 3.2. Shannon Diversity Exhibited Variation across Different Urbanization Gradients

With increasing sample size, the rarefaction and extrapolation sampling curve (Figure 3) flattened out. Since the number of butterfly species had reached its saturation, the minimal marginal contribution of extra butterfly individuals to the emergence of new species suggested that the data set as a whole was reliable. According to the GLM results, urbanization exerted a substantial influence on the Shannon diversity (*p* = 0.003) and abundance (*p* = 0.007) of all butterfly species but not on the richness (*p* = 0.241) of the butterfly community as a whole (Table 3). According to the findings of the ANOVA post hoc test, the overall butterfly community’s variety varied in two ways: urban vs. peri-urban (*p* = 0.025) and urban vs. suburban (*p* = 0.003). Between urban and peri-urban areas and between urban and suburban areas, there were differences in the total butterfly community richness (*p* = 0.001 and *p* = 0.005, respectively) (Figure 4). The variety and richness of the Lycaenidae of butterflies were similar across all levels of urbanization (Figure 5). The Shannon diversity of butterflies of the Pieridae differed significantly between urban and peri-urban (*p* = 0.004) and between urban and suburban (*p* = 0.001). In addition, there were notable differences in the abundance of the Pieridae between urban gradients, including urban vs. peri-urban (*p* = 0.001) and urban vs. suburban (*p* = 0.007) (Figure 5). The Papilionidae differed in Shannon diversity between urban and peri-urban (*p* = 0.027) and in richness between urban and peri-urban (*p* = 0.040) and urban and suburban (*p* = 0.031) (Figure 6). The diversity and richness of the Nymphalidae were similar across all urban gradients (Figure 6).

### 3.3. β-Diversity Exhibited Variation across Different Urbanization Gradients

The Bray–Curtis function was utilized to compute and order distances between overall butterfly communities and different butterfly families in urban parks at different levels of urbanization. The method tests whether differences between butterfly families were significantly greater than differences within families. In this case, ANOSIM similarity analysis was applied to the non-parametric test. The outcomes obtained from 999 permutations revealed that overall butterfly communities in urban parks differed between groups at different levels of urbanization (*p* = 0.020) and that the stress value of 0.142 (stress < 0.2) for butterfly communities was within a reasonable range. The strength of the repetition of the samples was determined by the distance between sample points within a group of NMDS analysis plots, and it shows that butterfly communities were more repeatable in urban regions and less repeatable in peri-urban and suburban locations (Figure 7).

### 3.4. Familial Role and Indicator Species

The “lm” function was used to examine the association between species richness and the Shannon diversity of butterfly families in urban parks under three different levels of urbanization. Due to the low number of species observed, Nymphalidae was excluded from the familial role analysis. Findings revealed a substantial and positive correlation between species richness and the Shannon diversity of the Lycaenidae within all three urban, peri-urban, and suburban areas (*p* = 0.005, 0.032, and 0.001, respectively). The same associations were observed for the Pieridae in the urban areas (*p* = 0.005), as well as in the peri-urban and suburban groups (*p*-values less than 0.001). Additionally, for the Papilionidae, we also discovered a highly significant positive association between Shannon diversity and species richness in both peri-urban and suburban areas (*p*-values less than 0.001) and urban areas (*p* = 0.007) (Table 4). Few indicator species were found in urban parks. *Lampides boeticus* and *Pieris rapae* were the most prevalent butterflies in peri-urban and suburban parks, respectively (IndVal = 0.546, *p* = 0.036 and IndVal = 0.475, *p* = 0.049, respectively) (Table 5). The other two urbanization types, however, lacked any notable indicator species.

## 4. Discussion

### 4.1. Urbanization Affects Butterfly Diversity in Urban Parks

According to the findings, the degree of urbanization significantly affected the species variety and richness of all butterfly species, but it had no significant impact on the community’s abundance. Peri-urban parks exhibited the highest species richness and Shannon diversity as compared to the other two urbanization levels, indicating that urban parks with moderate urbanization intensity maintained a high species diversity. However, Blair employed butterflies as indicators in urban environmental monitoring and discovered a consistent rising trend of butterfly indices from urban areas to natural areas. Then, he designed appropriate habitats to improve the conservation and restoration of butterfly diversity [62,63]. Interestingly, on the contrary, it has also been demonstrated that areas with higher levels of urbanization allowed for faster species growth and better invasion resistance, leading to a different species composition from other areas [27]. For instance, a sizable percentage of *Pieris rapae*, an indicator species of urban type, in Liming Lake Park, was discovered in the most urbanized urban park. The butterfly might be an omnivorous species that visited a variety of nectar plants with high nectar content if nectar plants with high nectar content were scarce [64]. It might be that crucifers (*Pieris rapae* host) were more common, so they were less affected by the level of urbanization. We found the lowest abundance of urban butterflies with only eight species, illustrating the adverse impact of urbanization on species richness even when the overall number of butterfly species did not respond significantly to urbanization. Similar to the findings of Olga Tzortzakaki (2019) [65] on the richness and number of butterfly species, dense Mediterranean cities with peri-urban areas outperform two other areas with higher construction densities.

### 4.2. Urbanization Affects Butterfly Families in Urban Parks

Hesperiidae was not detected in the research conducted for this study, likely due to their highly specific habits [66], and the high-intensity urban park space may not provide suitable conditions for their survival. In urban parks, urbanization had a substantial impact on the Shannon diversity (*p* = 0.011), abundance (*p* = 0.001), and richness (*p* = 0.002) of the Pieridae. The quantity and abundance of Pieridae species were highest in suburban parks, then peri-urban parks, and finally, urban parks. This is in line with the findings of Han Dan (2021) [67], who claimed that UZ5 (i.e., the least urbanized areas and urban fringe areas) had a much greater butterfly variety than other urban areas. Adults of the Pieridae are nectar feeders, and they prefer to land on the edge of shallow water to drink [68]. The primary reason for the considerable impact of urbanization on the Pieridae was that the majority of natural water bodies in suburban parks could provide them with shallow beaches to absorb water. Urbanization has little impact on the diversity distribution of the Lycaenidae, Papilionidae, and Nymphalidae compared to the Pieridae. Only *Lampides boeticus* and *Taraka hamada* were examined in the current study, which belongs to a small family of butterflies known as Lycaenidae [69]. Since both species were present in every urban park, urbanization had less impact on the spread of the Lycaenidae. The Papilionidae consumes animal excretion sap in addition to nectar from flowers [70], showing the diversity of its feeding sources. Therefore, the degree of urbanization has no impact on the Papilionidae’s ability to survive. The majority of Nymphalidae are energetic flyers, which makes it easier for them to access a large region for feeding [71,72,73]. Because of their wider range of movement, the variety of Nymphalidae may have been less affected by urbanization.

### 4.3. Urbanization Affects Butterfly Familial Roles and Indicator Species in Urban Parks

Among the three types of urbanization, there were differences in the familial variety of butterfly species found in urban parks. The Lycaenidae, Pieridae, and Papilionidae species richness and Shannon diversity were significantly and positively associated with urban gradients. However, in less urbanized areas, there was a highly substantial and positive correlation between the Pieridae and Papilionidae’s species richness and Shannon diversity. According to this, there might be more suitable habitats in peri-urban areas, which was consistent with the findings of Kong-Wah Sing (2019) [74]. Finally, our research showed that two butterflies, *Lampides boeticus* (peri-urban) and *Pieris rapae* (peri-urban), could be utilized to monitor urbanization changes in parks. The Indval index describes the frequency and number of each species. However, both indicator species were present in peri-urban parks. Due to the low frequency of individual butterfly species in urban and suburban parks, no relevant indicator species could be found there. These two indicator species belonged to the Pieridae and Lycaenidae, respectively, suggesting that the quantity of these community species in the suburban park is insufficient to convey their uniqueness. *Pieris rapae* is not particularly dependent on specific habitats, so it is able to adapt to urban environments [12,63] and can adapt to most urban environments, which are considered an urban-adapted species [75]. *Lampides boeticus* can mitigate the harmful effects of higher temperatures by expanding their habitats, but most butterfly species cannot do this [76]. This could be one explanation for why they occur more frequently in peri-urban parks with a moderate level of urbanization.

### 4.4. Insights for Enhancing Butterfly Diversity

The quality of green spaces is a crucial factor in butterfly survival [77]. Providing suitable natural or semi-natural habitats is essential for butterflies [78], as they require resources for both larval and adult stages [79]. Numerous studies have demonstrated a strong positive correlation between butterfly diversity and the availability of native plant resources in urban green spaces [80]. Planting specific host plants in parks has also been found to be effective in enhancing butterfly species richness [81]. There is little variation in pollution across the urbanization gradient, but parks that are more urbanized are usually managed more intensively, which can have a negative impact on butterfly diversity [82]. High management intensity may affect the dominance of graminoid and woody vegetation, potentially leading to a decrease in butterfly abundance [83]. Frequent mowing has been shown to lead to a rapid collapse of butterfly populations [84], so moderate management intensity is more conducive to building suitable habitats for butterflies. There are inherent problems in urban habitats, such as limited land availability and small areas, which increase the risk of species extinction. However, this risk can be mitigated by creating urban green corridors [85] and increasing their number in cities, which enhances butterfly diversity.

### 4.5. Limitations of This Study and Possible Developments

At each level of urbanization, urban parks were chosen for this study using a random sampling technique, which may have skewed the results. Future studies can increase the number of parks in various urban gradients to obtain more comprehensive butterfly study data. The distribution characteristics of butterflies in urban parks at various levels of urbanization were investigated in this study. Nevertheless, it has already been established that landscape and spatial elements have an impact on the richness of urban butterfly communities [86]. Future research into the impact of butterfly distribution at the patch scale may offer a theoretical direction for the development of green space in urban parks. Additionally, several studies have demonstrated that land cover [65], impermeable surfaces [87], and building density [88,89] might all have an influence on urban butterflies. Studies of correlation with these variables can offer empirical evidence for creating a healthy urban habitat environment. The diversity distribution of butterfly communities in other seasons of the year can be studied to determine the distribution patterns of butterfly diversity seasonally, which in turn can help build a more harmonious urban green space ecosystem. It is worth noting that this study only looked at the diverse distribution of butterfly species in Fuzhou city parks in the summer.

## 5. Conclusions

In this study, we investigated how different levels of urbanization in Fuzhou affected butterfly communities in urban parks. There were four conclusions found.

(1) In the most urbanized urban parks, the diversity and abundance of butterfly communities were reduced. We also observed that although urbanization had no discernible impact on the composition of other butterfly species, it had a considerable impact on the diversity, richness, and abundance of the Pieridae in urban parks. (2) Urban parks that are less densely populated may have more biological niches, hence supporting a greater variety and number of species. (3) Peri-urban parks had better butterfly composition and abundance than urban regions, suggesting that moderately urbanized areas could support more butterfly diversity. In order to foster the diversity of butterfly communities, we emphasize the importance of effective park conservation in urban areas with varying levels of urbanization. (4) We showed that the indicator species *Lampides boeticus* and *Pieris rapae* could be used to track the effects of urbanization in parks. To promote species diversity, urban parks should prioritize the quality of the vegetative habitat in addition to optimizing the ecological services provided by butterflies. Studies on host plants, nectar plants, and other plants may be involved. Our study contributes to management strategies for urban parks and programs for biodiversity restoration, which can lessen the detrimental effects caused by urbanization.

## Figures and Tables

**Figure 1 animals-13-01775-f001:**
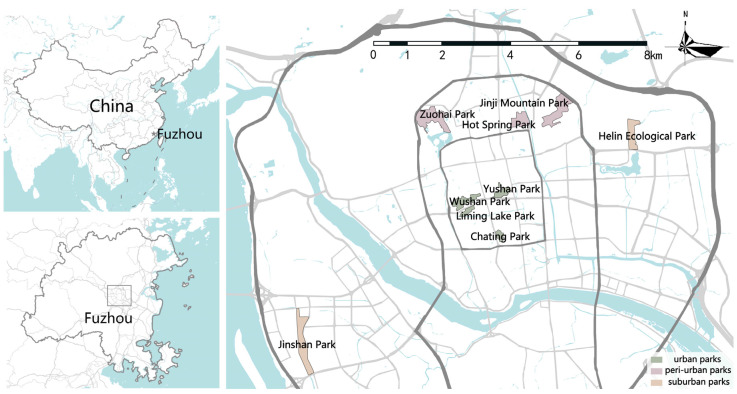
The study area is located in Fuzhou and includes nine urban parks. The Fuzhou high-speed ring road is depicted by a gray line, with the First Ring Road, Second Ring Road, and Third Ring Road situated from the inside to the outside, respectively.

**Figure 2 animals-13-01775-f002:**
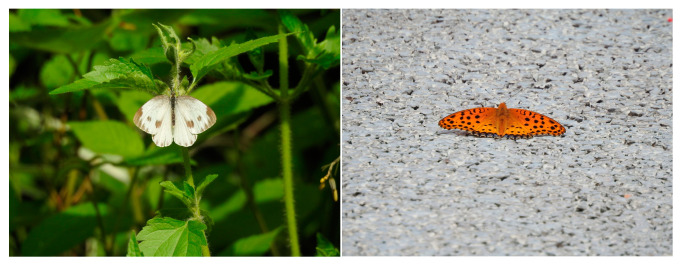
Some photos of the butterflies; on the left is the *Pieris rapae* and on the right is the *Argynnis hyperbius*.

**Figure 3 animals-13-01775-f003:**
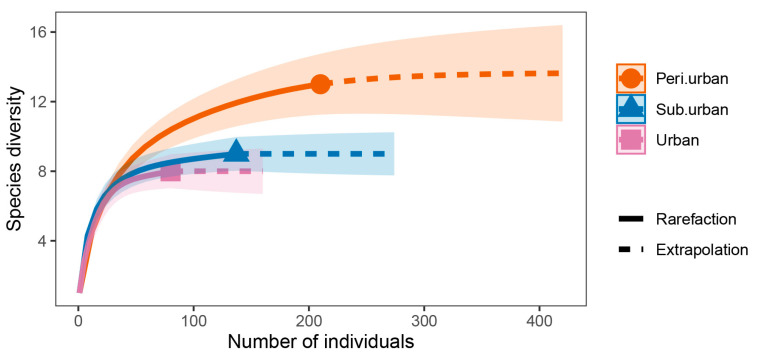
A comparison of interpolation (rarefaction) and extrapolation methods for estimating species diversity within individuals across urban parks of various urbanization types.

**Figure 4 animals-13-01775-f004:**
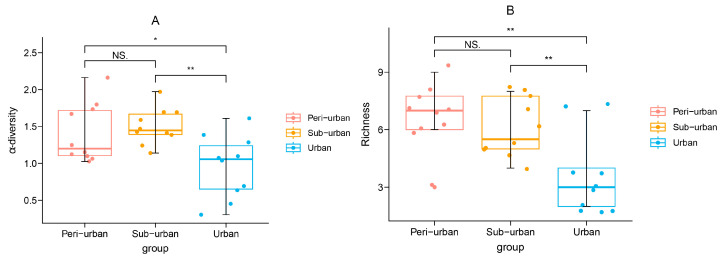
Box plots of overall Shannon diversity (**A**) and richness (**B**) of butterflies in urban parks of different urbanization types showing the range of data on median values, with leaving circles representing outliers and means represented by filled circles. Significance levels are denoted by asterisks (NS. = not significant; * = *p* < 0.05; ** = *p* < 0.01).

**Figure 5 animals-13-01775-f005:**
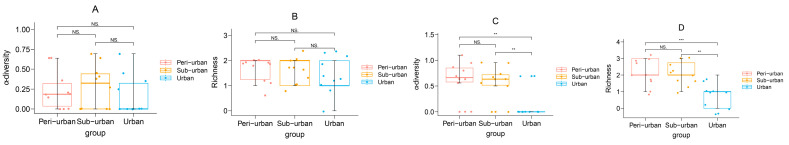
Box plots of Shannon diversity (**A**) and richness (**B**) of butterfly communities of the Lycaenidae in urban parks of different urbanization types and Shannon diversity (**C**) and richness (**D**) of butterfly communities of the Pieridae showing the range of data on median values, with leaving circles representing outliers and means represented by filled circles. Significance levels are denoted by asterisks (NS. = not significant; ** = *p* < 0.01; *** = *p* < 0.001).

**Figure 6 animals-13-01775-f006:**
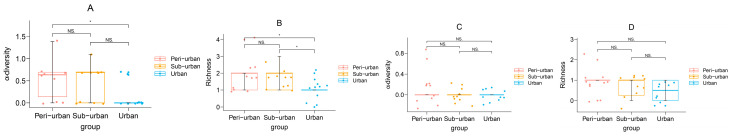
Box plots of Shannon diversity (**A**) and richness (**B**) of butterfly communities of the Papilionidae and Shannon diversity (**C**) and richness (**D**) of butterfly communities of the Nymphalidae in urban parks of different urbanization types showing the range of data on median values, with leaving circles representing outliers and means represented by filled circles. Significance levels are denoted by asterisks (NS. = not significant; * = *p* < 0.05).

**Figure 7 animals-13-01775-f007:**
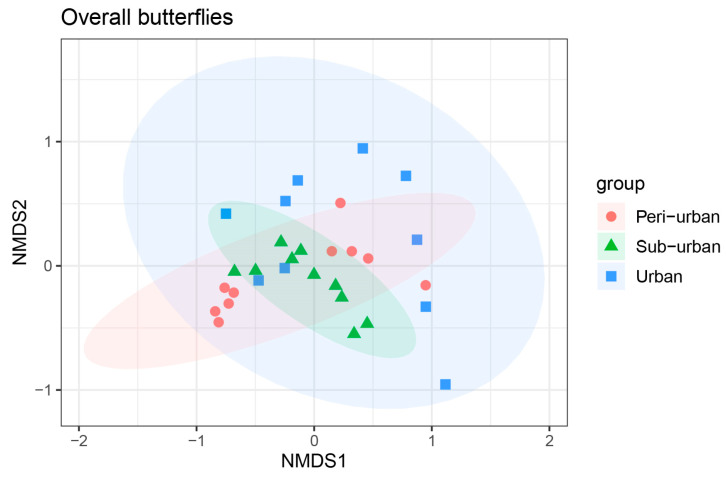
NMDS 2D maps of the overall butterflies class from 30 sampled transects (10 cities = blue squares, 10 peri-urban areas = red circles, 10 suburban areas = green triangles).

**Table 1 animals-13-01775-t001:** Scientific names and families of butterflies in different urban gradients, number of occurrences, and frequency. Frequency denotes the occurrence rate of butterfly species along the sample line within the urban gradient.

Type of Urbanization	Butterfly Family	Scientific Name	The Number of Individuals per Species	Frequency
Urban	Lycaenidae	*Lampides boeticus (Linnaeus, 1767)*	48	9
	Lycaenidae	*Taraka hamada (Druce, 1875)*	4	4
	Pieridae	*Pieris rapae (Linnaeus, 1758)*	6	4
	Pieridae	*Catopsilia pomona (Fabricius, 1775)*	6	4
	Papilionidae	*Papilio polytes (Linnaeus, 1758)*	5	4
	Papilionidae	*Graphium sarpedon (Linnaeus, 1758)*	5	5
	Papilionidae	*Papilio protenor (Cramer, 1775)*	1	1
	Nymphalidae	*Argyreus hyperbius (Linnaeus, 1763)*	5	5
Peri-urban	Lycaenidae	*Lampides boeticus (Linnaeus, 1767)*	124	10
	Lycaenidae	*Taraka hamada (Druce, 1875)*	9	7
	Pieridae	*Pieris rapae (Linnaeus, 1758)*	28	10
	Pieridae	*Catopsilia pomona (Fabricius, 1775)*	10	8
	Pieridae	*Eurema hecabe (Linnaeus, 1758)*	4	4
	Papilionidae	*Papilio polytes (Linnaeus, 1758)*	13	7
	Papilionidae	*Graphium sarpedon (Linnaeus, 1758)*	8	7
	Papilionidae	*Papilio protenor (Cramer, 1775)*	1	1
	Papilionidae	*Papilio memnon (Linnaeus, 1758)*	2	2
	Papilionidae	*Papilio helenus (Linnaeus, 1758)*	2	1
	Nymphalidae	*Argyreus hyperbius (Linnaeus, 1763)*	6	6
	Nymphalidae	*Vanessa indica (Herbst, 1794)*	1	1
	Nymphalidae	*Neptis sappho (Pallas, 1771)*	2	2
Suburban	Lycaenidae	*Lampides boeticus (Linnaeus, 1767)*	55	10
	Lycaenidae	*Taraka hamada (Druce, 1875)*	6	6
	Pieridae	*Pieris rapae (Linnaeus, 1758)*	25	10
	Pieridae	*Catopsilia pomona (Fabricius, 1775)*	22	8
	Pieridae	*Eurema hecabe (Linnaeus, 1758)*	3	3
	Papilionidae	*Papilio polytes (Linnaeus, 1758)*	10	10
	Papilionidae	*Graphium sarpedon (Linnaeus, 1758)*	6	6
	Papilionidae	*Papilio protenor (Cramer, 1775)*	1	1
	Nymphalidae	*Argyreus hyperbius (Linnaeus, 1763)*	9	7

**Table 2 animals-13-01775-t002:** Distribution of Shannon diversity, richness, abundance, and Chao1 index of butterflies overall and butterfly families in urban parks under three urbanization gradients.

Type of Urbanization	Grouping Variable	Parameter			
		Shannon Diversity	Richness	Abundance	Chao1
Urban	Butterflies overall Lycaenidae Pieridae Papilionidae Nymphalidae	0.958 0.174 0.139 0.133 0	8 2 2 3 1	80 52 12 11 5	5.23 1.40 0.80 1.10 0.50
Peri-urban	Butterflies overall Lycaenidae Pieridae Papilionidae Nymphalidae	1.408 0.235 0.625 0.53 0.069	13 2 3 5 3	210 133 42 26 9	13.18 1.70 2.80 2.80 1.00
Suburban	Butterflies overall Lycaenidae Pieridae Papilionidae Nymphalidae	1.502 0.284 0.565 0.456 0	9 2 3 3 1	137 61 50 17 9	10.95 1.70 2.30 2.50 0.70
Overall	Butterflies overall Lycaenidae Pieridae Papilionidae Nymphalidae	1.298 0.231 0.443 0.373 0.023	13 2 3 5 3	427 246 104 54 23	9.78 1.60 1.97 2.13 0.73

**Table 3 animals-13-01775-t003:** The GLM test results revealed the effect of three levels of urbanization on Shannon diversity in butterfly communities (* = *p* < 0.05).

Variable	Parameter	Estimate	Std. Error	*p*	df
All butterflies	Shannon diversity Abundance Richness	1.033 0.017 0.187	0.312 0.014 0.064	0.003 * 0.241 0.007 *	28
Lycaenidae	Shannon diversity Abundance Richness	0.609 0.004 0.317	0.618 0.019 0.268	0.333 0.815 0.247	28
Pieridae	Shannon diversity Abundance Richness	0.974 0.183 0.459	0.360 0.047 0.133	0.011 * 0.001 * 0.002 *	28
Papilionidae	Shannon diversity Abundance Richness	0.673 0.128 0.359	0.364 0.121 0.179	0.075 0.299 0.055	28
Nymphalidae	Shannon diversity Abundance Richness	−0.004 0.352 0.241	1.240 0.242 0.290	0.997 0.157 0.413	28

**Table 4 animals-13-01775-t004:** The linear regression models examine the relationship between Shannon diversity and species richness of butterfly families across varying urbanization levels in urban parks. The abbreviations in the table are defined as follows: LY is Shannon diversity of the Lycaenidae, PI is Shannon diversity of the Pieridae, PA is Shannon diversity of the Papilionidae, and SR is species richness.

Node	Butterfly Families	Type of Urbanization	Model	F	R^2^	Sig.
1	Lycaenidae	Urban Peri-urban Sub-urban	LY = −0.212 + 0.297 SR LY = −0.336 + 0.336 SR LY = −0.473 + 0.473 SR	14.260 6.737 33.140	0.641 0.457 0.806	0.005 0.032 0.001
2						
3						
4 5 6	Pieridae	Urban Peri-urban Sub-urban	PI = −0.099 + 0.297 SR PI = −0.345 + 0.441 SR PI = −0.345 + 0.297 SR	14.400 71.490 61.380	0.643 0.899 0.885	0.005 0.000 0.000
7 8 9	Papilionidae	Urban Peri-urban Sub-urban	PA = −0.199 + 0.332 SR PA = −0.245 + 0.430 SR PA = −0.567 + 0.602 SR	13.250 41.370 294.400	0.624 0.838 0.974	0.007 0.000 0.000

**Table 5 animals-13-01775-t005:** Butterfly species reflecting the different types of urbanization in urban parks. We have indicated in the table the species with the highest correlation, the corresponding indicator species value (IndVal), and the statistical significance of the correlation (*p*-value, * = *p* < 0.05;).

Indicator Species	Type of Urbanization	IndVal	Frequency	*p*-Value
*Lampides boeticus*	Peri-urban	0.546	29	0.036 *
*Pieris rapae*	Sub-urban	0.475	24	0.049 *

## Data Availability

All images in the text were drawn by the author. The data used to support the findings of this study are available from the corresponding author upon request.

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
