# Peer review of "Butterfly Communities Vary under Different Urbanization Types in City Parks"

_animals, 2023, doi:10.3390/ani13111775_

Round 1
Reviewer 1 Report
With the growth of human population and urbanisation, ecologists and conservation biologists are increasingly interested in urban ecosystems, because some natural phenomena exist even there, and because urban locations are those where many people, in fact, experience nature most frequently.
For these and other reason, I personally much welcome this paper, investigating butterfly diversity in urban parks in a Chinese tropical metropolis, Fuzhou, showed - expectably - a decline of butterfly diversity from city cener towards peri-urban setting, and use this information to inform city planners.
The paper is of particular interest, because similar studies had been done in Europe, North America, and elsewhere in a past, and rapidly growing nations such as China provide beg for replication of such studies in settings that are more diverse biogeographically, and ... well, have more and bigger cities.
So much being said, the paper suffers two serios drawbacks. One is some naivete of presentation, perhaps explicable by low experience of the authors, which I will indicate in "Minor remarks", below. These issues can be rather easily corrected during revision round, and I am sure the authors will perform the job perfectly.
The second, and far more serious, is considering "Diversity of families" as proxy for functional diversity (lines 258 - 263 in Material and methods). Although there is some point (Lycaenidae tend to be small, Papilionieae large...), the problems here are, that functional traps across the families greatly overlap (think about size of Pieridae vs. Nymphalidae, but also such issues as host plant use, flight performance, numbers of generations per year...). Moreover, the families are not equivalent, Nymphalidae being by far the species- and functionally richest, which will undobtedly reveal some statistical issues if dissected in details.
It the authors insist on analysing functional diversity, they should tabelarize some tangible traits of individuals species (the most easily obtained ones should be wingspan, host plant growth form - if known, flight period duration in Southern China, size of global range, perhaps preferred adult feeding mode, and perhaps vertical use of habitat - ground, shrubs, tree crowns), but I am not sure how is this possible for their sample of species. Then, they should proceed with these "functions", as they did with families.
But perhaps, they might leave this issue for a next paper, and either omit the families analysis completely, or replace by "familial diversity", or something in this line. They of course may discuss that the families are somehow proxies to functional diversity, but only up to a point.
Following my advice will probably lead to shortening of the paper (unless the authors produce a "true" functional richness analyses). One table (table 4) will get away for good, although Table 1 will be expanded. I ask the authors not to feel "sorry" for the deleted parts - in scientific writing, it is sometimes "the shorter, the better", and reducing size of a paper usually helps to rise interest of readers.
And now, all the minor/stylistic remarks.
1st para. I would slightly alter the entire framing, from claims that urban ecoystems are "damaged" to, perhaps, that they are heavily altered, completely transformed, or simply "novel" ecosystems (compared to what was there centuries before). In reality, many cities across the World are actually undergoing various biodiversity friendly "greening" actions, and may in fact be much less "damaged", than they were half-century ago. The term "transformed" covers both situations, you can transform for either worse or better.
Perhaps most suitably, I recommend citing some of the "classics" of urban conservation, nicely summarizing the rationale for studies such as yours,
Miller, JR, and Hobbs RJ, 2002, Conservation biology 16, 330-337.
Plus, although I understand that the patterns reported from China alone suffice to illustrate your points, consider adding a non-Chinese example or two, what about this one:
Ferenc, M., et al., 2017, Global Ecology and Biogeography, 23(4), pp.479-489.
line 50: "many foreign scholars" - at this moment, a reader does not yet know that your study focuses of China, what about "scholars across the World", "scholars across continents" or something similar.
line 78: " particularly in the south-eastern region of China" - " particularly in the south-eastern tropical region of China" (or perhaps subtropical, I am not checking now, but the point is to guide a reader a bit, without necessity of checking a climate map, how the region looks like).
Material and Methods
somewhere around line 98: Add number of inhabitants, perhaps population density, and possibly a line, not more, about history of the city, which would indicate how rapidly it has grown. Something like, "It is historical city, dating back to centuries before common era, and underwent rapid development during the 20th+21st centuries, rising from population size from xxx in 1900s to current 7.4 millions" (I took this from Wikipedia, I am sure you will have better sources).
around 100-116 + 122-151 (plus the accompanying figure)
This lengthy part basically describes the conditions of the parks and their selection (plus growth of the city, the Ring roads, etc.). Although interesting, it must me shortened and reorganized. It appears absurd, while reading, that you first select all the parks, describe them with all the details (waterworks, gardening details, etc.), and then go to selection.
My suggestion: There is xxx parks in total (xxx stands for a number); their level of urbanisation is well proxied by their position in relation to the ring roads, and we selected the following, belonging to such-and-such urbanisation classes. Then, as a next paragraph, you may give some details of the parks, including area, etc. But the selection procedure must go first.
And regarding the map, I would indicate the urbanisation levels / classification directly to the map, e.g., by different colours.
line 153: "A 100-meter sampling line" is grammatically incorrect, as there were several such lines in some of the parks. Rethink. Perhaps, "The sampling proceeded along 100-meter long sampling lines put haphazardly..."
lines 164-5: "3 times in the same season to get better results". Nope, you did it "to include main phenology aspects", please, change.
lines 177-9: Do not name the identification literature in the main text (you are naming the other references either). Just write st. like, "We used standard identification literature [citations]"
180- 190: Here, you are basically reiterating family classification of butterflies, which is truly not necessary, unless related to the story. Entomologists know these things. Also, 190-193 reiterate what was already said in Introduction. I recommend deleting, or radically reducing, this paragraph.
232-235: That "community ecologists believe" is not an argument, but faith. Re-frame the argument in more scientific terms, e.g., "We used The Bray-Curtis dissimilarity coefficient, which xxx (and name its advantages for your study; referring to Ricotta and Podani, 2017, Ecological complexity 31, 201-205 may help).
Table 1. The table shows very little, for its size. You need somewhere, perhaps here, present more details, e.g, the number of individuals per species, frequency (= on how many transects, in how many parks...), etc. You might find inspiration in Konvicka and Kadlec, 2011, European Journal of Entomology 108, 219-229.
Also, and critically. Species names must be gvien in full, i.e. with authority and year of descirption, when mentioned for the first time. So, for instance, the full name of L. boeticus is "Lampides boetics (Linnaeus, 1767)".
line 286: Replace "As the population grew" by "With increasing sample size".
lines 345-356, and Table 4
Shannon diversity index will always increase with species richness, as it is derived from it, and so, the whole richness-Shannon part is superflous, and should be deleted (together with associated parts in Discussion).
The indic. species section (356-9) is OK.
Discussion
Somewhere in Discussion, the authors may mention some of the studies attempting to recommend, how to improve urban habitats for insects. There are many such recent papers, I understand that a complete review would go beyond the scope of this paper, but a few exapmples may deserve mentions, e.g.,
Konvicka, M. and Kadlec, T., 2011. European Journal of Entomology, 108, 219-229.
Klaus, VH, 2013, Restoration Ecology 21, 665-669.
Horak, J.,et al., 2022, Urban Forestry & Urban Greening, 73, p.127609.
Breed, C.A., et al. 2022. Land, 11(8), p.1171.
Minor editing, focusing not on grammar, which is OK, but on scientific terminology and a bit more "professional" language would help.
Reviewer 2 Report
A nice paper with useful information on the impact of urbanization on butterflies.
Suggestions:
1. Survey was based on a modified Pollard walk. This should be mentioned in the text and referenced in the references .
2. Line 50: replace 'scholars' with 'scientists'
3. All scientific names need italicizing (ie in Figs and Tables)
4. In Table 1, organize better. For example the word 'family' is redundant on most lines.
5. Fig. 7 'Butterflys' should be 'butterflies'
6. Line 384 'gived' is not a word.
7. There should be some discussion on the possible impacts of pollution and butterfly host plants, affecting the observed abundance and diversity. Are the urban areas more polluted? Or is there little difference over the entire area? What is the host plant availability for the 13 species? It would be useful to show this. For example, perhaps crucifers (P rapae host) are more common in highly urban areas, aiding abundance of this species?
English quality is okay. However, there are occasional errors that need fixing. eg 'gived' (line 384)
Reviewer 3 Report
Lin et al.
Butterfly communities vary…
General
An interesting study that certainly merits publication. The ms is well-written, and I have but a few comments.
A recurring thing is that the term "family" is repeated multiple times in connection with the latin name of the (butterfly) family, which is superfluous, unusual or even peculiar.
For ex, the term "Lycaenidae" refers to a butterfly family.
See all Tables and some occurrences in the text.
The study is complete as it is, but attempts to somehow document vegetation type, abundance of plant species etc. would have been useful.
Specific
176. specimensmaking = identification
183. …
282. delete "of the butterfly" in Table header
Caption Figure 3. Hard to understand, incl term "reasonable"
311. Table 3 "All butterflies" (y = ies)
401. shallow terrace to land = rephrase
Overall ok, language and style are good, but some technical terms could be improved.
Round 2
Reviewer 1 Report
The authors did tremendous work during the revision, but there is still some work needed.
My first question, related to the butterfly families, which I did not notice in the previous revision:
What about Hesperidae?!?
They are also butterflies - and they have many representatives in your region. Were they missing in the parks, or did you omit them, e.g., because of troubles with identification? In the latter case, this is a problem, but you may self-critically admit it, and be forgiven. In the former case, an absence of hesperids would require a comment in Discussion, otherwise, every reader will ask similar questions.
Second, scientific names of the butterflies. When I asked you to give them in full, I meant - give them in full at the first mention, not throughout the whole article. Table 1 seems to be a good place to do it, so, list the full names there, and please, nowhere else.
Minor remarks
lines 192-3: literatures => literature
194-8: The part about classification into families still sounds naively, as every entomologist, including (a believe) amateurs, knows which families form the monophyletic superfamily Papilionoidea, i.e., butterflies.
What about changing the whole paragraph as,
"We use classification into families, following, e.g., [36,37], considering that members of the main families share some life history features. Lycaenidae range in size..."
line 224: "in an ecosystem" change to "in a sample" (nobody ever has data for entire ecosystem, all we have are samples).
Table 1. I am glad that you expanded the information presented, but with this, the caption should be expanded as well. What is "frequency" - is it the number of parks, or visits, when a species was seen?
428-447: the paragraph not in italics
/
